# A Brief Overview of Medical Fiber Optic Biosensors and Techniques in the Modification for Enhanced Sensing Ability

**DOI:** 10.3390/diagnostics9010023

**Published:** 2019-02-27

**Authors:** Shannon E. Mowbray, Amir M. Amiri

**Affiliations:** Department of Biomedical Engineering, Widener University, Chester, PA 19013, USA; semowbray@widener.edu

**Keywords:** biosensors, fiber optic sensors, immunosensors

## Abstract

In this paper, we provide a brief overview of fiber optic biosensors for use in MedTech, specifically to aid in the diagnoses and treatment of those with chronic medical conditions. Fiber optic cables as components of biological sensors make them especially effective in biological systems that may require ultra-sensitive detection of low-frequency signals in hard to reach areas. This systematic review focuses on the differentiating factors of fiber-optic biosensors, which are tailored to apply the sensor to specific health needs. The main components of FOBS (fiber optic biosensors) such as biosensing elements, fiber optic cables, optical element enhancements, transducers, sensing strategies, photodetectors, and signal processing are covered in detail by showcasing the recent developments in modifications to these components. This paper pays particular attention to the alterations made in biosensing elements including pH elements, enzymatic elements, as well as those sensors utilizing antibodies and whole-cell bacteria. This paper reviews and discusses several published examples in the research stage of development to give the reader an overall scope of the field. The need for research on biosensing equipment is increasing, as the number of individuals with chronic diseases and the geriatric population require more effective, accurate, and mobile sensing ability and reduced invasiveness. FOBS offer a sensing solution that is accurate, tailorable to almost any clinical need, has abundant and relatively cheap material requirements, and a well-established technological base in fiber optic technology. This small price tag and large market potential make FOBS a desirable research area.

## 1. Introduction

According to a report by Global Market Insights, the United States biosensing market will exhibit a 7% increase by 2024, reaching a potential total of $30 billion [1]. According to the Centers for Disease Control and Prevention, approximately 40% of the United States population is afflicted with a chronic disease, reaching a total of over 133 million individuals [2]. Examples of chronic illnesses that are becoming more prominent and are pushing this industry forward are diabetes, heart disease, and cancer. As stated in a research paper written by Devol and others out of the Milken Institute in California in 2007, seven chronic diseases (including diabetes, cancer, heart disease, and stroke) have a total impact on the United States economy in treatment costs and lost output of $1.3 trillion every year. This staggering statistic is expected to jump to a projected $4.2 trillion by 2023 [3]. An increase in the availability and diversity of sensing technology will greatly affect the ability to monitor and treat chronic diseases.

There are three main categories of biological sensors in which fiber optics are applicable. Those categories are imaging sensors, physical characteristic sensors, and biochemical sensors [4]. The latter being the least developed group at present. Fiber optic cables are commonly used as extensions of sensor-amplifier systems to extend their sensitivity to signals and increase the range of the sensor. For example, fiber optic cables are well established in endoscopic imaging but are beginning to emerge for use in coronary imaging, oxygenation, as well as temperature and pressure detection. Fiber optic biosensors (FOBS) for medical applications can be used both invasively and non-invasively. These sensors are electrically safe and small enough to reach small or hard-to-reach areas of the body. FOBS have been used in applications such as the analysis of gases, tissue, or bodily fluids, as optrodes on the skin, as catheters, and as endoscopic tools [5].

Biosensors are a diverse category of sensors and can be differentiated not only by their mechanical components but by the biological sensing unit they utilize and several other specialization mechanisms. Although biosensors are widely used in the medical field today, at their origin they were simple, utilizing a small number of mechanical components, a far cry from the diverse and complex biosensors that are detailed in this paper. The first biosensor was developed by Updike and Hicks in 1967 to measure blood glucose levels using a natural human enzyme (glucose oxidase) and an electrochemical detector [6,7]. This technology has since been refined and is in use in compact glucose monitors to help treat and monitor diabetes around the world. Since then, biosensors have evolved to monitor more a more diverse array of diseases, become smaller and increasingly complex.

Biosensors can be broken down into several categories, first by transducer mechanism, and then by components, as presented in Figure 1. FOBS are differentiated from other biosensors by the optical-based transducer that uses absorbance, reflectance, luminescence, refractive index, and light scattering to alter the signal for processing [8].

An optical fiber is a rod formed from chemically treated glass or silicon that is heated to around 4000° F and drawn by gravity into tubes of varying diameters. The size of the tube is monitored and altered by a laser micrometer as the tube is drawn through several coating stages. This allows the achievement of specific refraction angles, light propagation properties and characteristics that stem from the coatings applied to the rod [6]. When a light signal interacts with the optical fiber in a sensor mechanism, the light is propagated down the core of the fiber where it interacts with the sensing element. The element then filters the light and sends it to the detector where it is further processed by other instrumentation. FOBS utilize a specific sensing element that gives these sensors their niche in the MedTech industry. This element is called a biosensing element. Examples include enzymes, molecular imprints, lectins, receptors, antibodies, and nucleic acids. The general flow of a signal as it propagates through a FOBS, as described above, is outlined in Figure 2.

Although all FOBS operate utilizing an optical based transducer, they are adaptable to many different applications. The differences in use arise through the modifications made to the components of the sensors. The general structure of FOBS is outlined in Figure 3 and shows the main categories of components that differentiate FOBS from each other, from other biosensors and that allow them to perform specific medical applications. The “optical enhancement” section of Figure 3 was included for completeness, but those listed components are not covered in detail in this paper because of the modifications made to those components do not affect the sensors ability to adapt to medical needs and applications. For example, different optical elements can be utilized to manipulate the light signal input to achieve better detection by the FOBS. This includes mirrors and beam splitters that are used to alter the pathway of the light beam to achieve a more potent signal to be propagated by the fibers in the sensor. Optical transducers (a component of all FOBS) link changes in light intensity to differences in mass or chemical concentration. Commonly used optical transducers are surface plasmon resonance (SPR) interferometers, resonators, and refractometers. FOBS can guide light of different wavelengths simultaneously, giving them the ability to monitor and deliver information on more than one function at a time. FOBS can be adjusted from the general structure (one component from each section in Figure 3 below) for fine-tuning, and adaptation to more sensitive sensing in the body. Each component must be chosen for the sensor after considering the specific aim of the sensor. For example, the depth of signal penetration, the distance the signal must propagate, naturally-occurring internal noise produced by the body and several other factors are considered as the sensor is constructed. A good example of this practice is the photodetector (a detector specifically designed to capture data from light photons). This instrumentation must be sensitive enough to achieve accurate results, while not producing noise that would distort the signal. Several types of photodetectors can be used in FOBS applications, depending on the signal that is being captured [6].

## 2. General Components and Modifications of FOBS

### 2.1. Fiber Optic Cables and Other Optical Elements

The optical fibers that make up the backbone of FOBS consist of two main parts, constructed of glass, silica, or plastic. The inner tube, called the core, is surrounded by a casing with a larger diameter, which is called the cladding. This common fiber optic structure is shown in Figure 4. The “jacket” in the figure refers to the optional coatings and insulating layers that can be included superior to the cladding. The two main parts can be constructed in one of two ways. The first allows the light signal to propagate in only one direction, straight down the core. The second way allows the light signal to propagate in many paths through the core. The difference in signal propagation is usually accomplished by coating the cladding. For example, a coating of germanium increases the refractive index. The light signal through an optical fiber breaks into two sections. One section becomes a standard, guided field through the core. The second becomes an evanescent field that decays exponentially to zero in the cladding. These two phenomena are shown in Figure 3 below. In biosensing, a portion of the cladding must be removed or compromised to allow the evanescent field to interact with the surroundings and allow the other instrumentation in the sensor to pick up the signal [9]. One strategy to expose the cladding is called tapering. Tapering can be performed on the core, cladding or both. An example of a tapered tip, occurring at the end of a fiber optic element, and a tapering of the cladding of a fiber optic element are illustrated in Figure 5. Depending on where the tapering occurs, differences in evanescent field magnitude and depth are seen. The evanescent field can also be altered by bending the fiber (resulting in a loss of light and increase in the field), changing the launch angle, increasing the wavelength (to increase the evanescent field depth) or a combination of multiple strategies [9].

As opposed to the standard glass fiber optic cables typically used for FOBS, a sensor has been developed to determine the concentration of cholesterol in human blood, in real time, using a plastic fiber optic cable. The sensor utilized a single fiber optic cable, a detector, a personal computer (PC) to interface and display the results, and a spectrometer to supply the light signal. The cable was further modified by scraping the sensing area by five scratches and altering it by 5 cm with a bending treatment. The scraping changes the way the light signal is propagated in the cable and increases the sensitivity to specific wavelengths of light. To test this sensor, several blood serum samples were measured using a light spectrometer to determine the specific wavelength that correlated with each standard concentration of cholesterol. The adjusted sensing area of the cable interacts with the blood serum directly [10]. Biconical, nonadiabatic tapered optical fiber design has been innovated for use in cell protein and DNA based biosensors. These sensors are of a lower cost than other FOBS and have a higher sensitivity [11]. Biconically tapered fibers consist of three regions, including a centered “waist” region that has a smaller diameter and acts as the sensing region [9]. This construction is shown in Figure 5. Regions a and c in Figure 5A represent the two unaltered ends of the fiber that surround the central “waist” region (denoted by c in the Figure 5). Tapered FOBS have several unique applications. Ferreira et al have utilized this technology, as well as, chemical etching on the fiber, to detect the growth of Escherichia coli [12]. The detection of whole cells for clinical measurements has been little researched. This is probably due to the difficulty in culturing human cells and the variability present even in one population (or type) of the cell for sensing purposes. However, tapered fiber optic cables have been used by the Chinese to detect hamster ovary cells [13].

### 2.2. Biorecognition Elements

The biosensing, or biorecognition element is affixed to the fiber optic cable and directly interacts with the sample analyte that is being analyzed or detected. This component of every FOBS is a unique factor that allows this area of sensing technology to have its diverse list of applications. The element acts as a catalyst for the detection and initial transmission of the signal. Most commonly, these catalytic FOBS are extrinsic sensors which guide the light signal to a separate sensing region. This is opposed to intrinsic sensors that force the light signal to remain within the core of the fiber to detect any changes in intensity, phase, polarization or wavelength that occur as the signal is propagated. In these extrinsic FOBS, the fiber optic cable transmits the light signal from a biorecognition element that is immobilized and attached to a free end of the fiber [14]. There are several types of biorecognition elements that are as diverse as the applications of FOBS themselves. Those covered in this paper are pH sensing, enzymatic biosensors, immunosensors, and bacterium-based biosensors.

#### 2.2.1. pH Biorecognition Elements

In the human body, the blood has an optimal pH range under which it operates most effectively. The human body is also sensitive to small adjustments in pH and the ability to accurately determine minute changes is achievable through the accuracy and sensitivity of fiber optic sensing systems. Several researchers from the University of Oviedo in Gijón, Spain developed an optical sensing system that utilized two sensing fiber optic cables, an inexpensive photodiode and a biosensing element that consisted of mercurochrome (a pH indicator) fixed to a gel matrix. After an initial blue light excitation, the fluorescence emitted by the pH indicator was then able to be detected by the photodiode. The second fiber optic cable detected the intensity of the initial blue activating signal. The ratio of these two signals provided the pH. The results showed that with increasing pH, the fluorescence also increased, in a linear fashion. These experiments require new gel to be made each time the experiment is conducted due to decomposition of the dye and long-term stability of the results could be improved with a screen printing of the immediate results [9]. Although research into fluorescence as a signal indicator for use with fiber optic sensors is becoming more common, uses of this technology for pH indication is rarer. In another application, a two-fiber optic cable system (miniaturized to a total diameter of 0.4 mm) has been developed to sense the color change of a pH indicator dye. This system is specialized to be accurate to 0.01 pH units within the range of 7.0–7.4 which directly aligns with the normal blood pH for humans [15]. There are currently no developed FOBS systems to directly measure the pH of biological fluids, as this example was tested in a non-biologic fluid. A clinically relevant FOBS with the aim to detect pH accurately in biological fluids would have to be developed and tested in living human specimens. This kind of research endeavor is expensive, harder to obtain the proper permissions for and has a longer overall process. There are also advantages to developing a sensor outside of this realm and proving its effectiveness before introducing it to clinical testing and applications.

#### 2.2.2. Enzymes as Biorecognition Elements

Enzymes are specialized proteins that catalyze biological chemical reactions. For use as a part of FOBS, they can either be present as part of a tissue sample or suspended in media as individuals after separation. These enzymes work in three unique ways. Option one involves the conversion of the sample into a detectable product by the enzyme. In the second option, the sample inhibits or activates the enzyme which then translates a signal. The final option produces a signal based on the changes made to the enzyme active site after the sample binds [16,17]. Enzymes as biorecognition elements are most commonly used in optical-based glucose meters aimed to determine the concentration of glucose in the blood. Developments in the 1990s updated this technology to include oxygen as a partner in the enzyme biorecognition element affixed to the optical fiber. In glucose solutions of a neutral pH (mimicking human bodily fluids) the sensor was able to detect oxygen consumption as a result of enzymatic oxidation and relate it to glucose concentration [18]. A more recent development in glucose monitoring, enzyme-based FOBS is in a microsensor that is inserted in the subcutaneous tissue of the patient. This microsensor also utilizes an oxygen optode (with immobilized enzyme glucose oxidase) fixed to two fiber optic cables. In this sensor, the difference in the pO_2_ between the two cables shows the oxygen level fluctuations and is translated into a glucose concentration signal. This method shows reduced sensitivity to glucose signal resolution but high sensitivity to low oxygen levels in the intercellular fluid [19]. Again, utilizing the same FOBS structure and enzyme to monitor glucose levels, Brown et al. have developed a microspherical FOBS glucose sensor utilizing an ultra-thin (12 nm) film on which the enzyme and fluorescent indicator are imbibed [20]. Other enzyme-based FOBS includes a sensor developed out of the University of Maryland School of Medicine in 1993 that used fluorescence-based signals, emitted by an inhibitor, that bonded to zinc ions on the active site of the enzyme to make measurements on the nanomolar scale. This technique’s instrumentation proved too fragile and of too low a sensitivity level to detect the low levels of zinc in areas such as the ocean, or the human internal biological environment [21]. Another enzyme-based FOBS has been developed for the detection of hydrogen peroxide. Future projections of this technology hope to detect levels of uric acid, D-amino acid, L-amino acid, glucose, cholesterol, and acetylcholine using the same immobilized enzyme. Oxidase and horseradish peroxidase (HRP) are immobilized on the bovine extracellular matrix. This type of sensor is ideal for detecting hydrogen peroxide to measure the reaction rates of substances that produce hydrogen peroxide under catalyzation [22]. Also, for the detection of hydrogen peroxide are FOBS utilizing peroxidase or xanthine oxidase (harvested from microbes) to aid in flow injection analysis of hydrogen peroxide. This system is being adapted for the detection of glutamate, lysine, xanthine, glucose, lactate, glutamine, ammonia, hypoxanthine, and phosphate [23].

#### 2.2.3. Immunosensors

Immunosensors can be described as FOBS whose biorecognition element is an antibody. Antibodies are proteins that are produced by lymphocytes as a part of the immune system. These proteins are variable and highly selective. They also require a small range of specific environmental characteristics to function properly, such as pH, lack of detergents, and/or definitive ion concentration gradients. Like enzymatic-based FOBS, the antibodies are immobilized on the sensor; however, they require an attachment agent. In some cases, the attachment agent is an amine or carboxyl group [6]. In other cases, the antibody can be immobilized using other techniques. The key to any antibody immobilization is the orientation of the antibody relative to the fluid phase it is suspended in. In the human body, antibodies work in conjunction with their complementary antigen. The mechanisms that define the use of these proteins in the body, are capitalized upon for their use in FOBS. The sensor can utilize an immobilized antibody that binds and interacts with its respective antigen to generate the signal, or the opposite phenomena, where the antigen is bound to the sensor. Andrade et al. modified a fiber optic cable by removing the cladding (leaving just the inner core) to induce interaction of the fiber with the environment using evanescent waves. The sensor analyzes the biological environment’s level of any antigen (Ag) by monitoring the concentration of fluorescent antibodies that are bound at the sensing site. The antibody molecules were immobilized on the surface of the silica-based fiber using a covalent bonding technique. The biorecognition molecule can also be modified to be an antigen and will bind its complement. If the binding sample is fluorescent, this signal can be used to perform an immunoassay. This sensor was tested for accurate use in human blood with sensitivity up to 10^−11^ moles per liter (M).The application of a fiber optic sensor to facilitate an immunoassay allows the test to be suitable for continuous measurements or those that need to be conducted in hard to reach areas. This application is continuing to expand and has since been applied to an insulin immunoassay and detection of absorbed immunoglobulins [24]. Applications of antibody immobilization on fiber optics for use as biorecognition elements were seen in the ability of a diagnostic assay completed using FOBS to detect levels of Protein C (PC), a key anticoagulant using a fluorescently tagged antibody for the protein. The system utilized a quartz fiber affixed with antibody ‘anti-PC’. To measure the concentration of PC in the sample, excitation light is applied through the fiber, and the fluorescence intensity is correlated with the PC concentration. At publication, the sensor had only been tested in a buffer solution but showed detection of concentration levels as low as 0.1 μg/mL. Future work hoped to include the automation of the system [25].

#### 2.2.4. Whole-Cell Biorecognition Elements

Microorganisms, or whole cell bacteria, are used as biorecognition elements because of the unique characteristics surround their metabolism. The consumption of oxygen or carbon dioxide by some microorganisms is a consistent phenomenon that makes it an ideal measurement criterion. Bacteria are also cheaper to purchase than enzymes or antibodies. These whole-cell biorecognition elements must be immobilized before attachment to the fiber optic cable, usually by nylon netting or membranes [6]. In comparison to enzymatic FOBS, sensors endowed with whole cell organisms are more selective in the samples they can bind meaning, they have a narrower range of perspective analytes than enzymatic or antibody-based biosensors. Common microorganisms that are used for optical signaling include those bacteria that produce a green fluorescent protein or bacterial luciferase (Lux) [26]. Three portable biosensing devices have been developed using bioluminescent cells to detect androgens, estrogens, and a lactose analogue, respectively. The cells were immobilized on the tapered fiber optic cable using a biocompatible matrix that made the biorecognition system viable for up to one month without replacement [27].

### 2.3. Transducers and Sensing Strategies

FOBS rely on optical based transducers. Optical transducers convert light signals into electrical signals that can be translated onto a display or another form to communicate the results. In FOBS, the optical fiber acts as the transducer. Commonly used signal generation strategies are surface plasmon resonance (SPR), fluorescence, interferometers, refractometers, and changes in absorption or refractive index.

#### 2.3.1. Surface Plasmon Resonance

SPR is the phenomenon that occurs as a result of light hitting a metal film (part of the instrumentation). SPR strategies excite and detect the movement of free electrons (surface plasmons) and record this information. As part of the instrumentation, the light is focused through a prism. The resonance angle that is produced as the light signal is absorbed can be measured and associated with molecular binding events as the shift in the SPR reflection intensity changes [28]. FOBS utilizing SPR are categorized based on the detection limit and the type of sample being tested. For example, long-range SPR can be applied in a bulk solution to test for a range of samples. SPR imaging can be implemented to detect specific proteins up to a detection limit of 1nm. Wavelength modulated SPR and flow injection SPR are especially effective in detecting DNA at varying detection limits [29]. A reusable FOBS has been developed to detect and measure DNA hybridization levels and DNA-protein interactions. The fibers are coated in a 50 nm gold layer. The sensor was evaluated using a DNA hybridization assay and was found to have a limit of detection of 2 nM [30].

#### 2.3.2. Interferometers

SPR, fluorescence, and changes in refractive index are more common in FOBS instrumentation and limited recent research is being conducted using interferometers as the prominent sensing strategy for the FOBS system. However, interferometers have a long history for use in conjunction with FOBS and helped to increase their use, beginning in the early 1900s [31]. The most widely vetted interferometer in FOBS applications is the Mach-Zehnder interferometer (MZI). In this system, a single laser sends light into a single waveguide. This tube then splits into a Y shape where one branch serves as the reference arm and the other has its cladding removed to serve as the sensing arm. As the two branches connect at the output, the intensity of the interference is measured. Several modifications to this system have occurred thereafter, including Young’s multichannel interferometer sensor and the Hartman interferometer sensor [29].

### 2.4. Photodetectors and related Biosensing Strategies

Sensing of the desired sample is integral to the proper operation of FOBS. Without the proper detection of the signal, the transducer cannot receive or transmit the desired data. Sensing in FOBS is dependent on one of two processes. The first is spectroscopy (fluorescence and absorption) and the second is evanescent wave interaction. This paper provides a quick overview of both strategies to enhance the reader’s understanding of the most recent developments in FOBS research.

#### 2.4.1. Spectroscopy

There are several kinds of spectroscopy that utilize optical components to detect and focus the radiation. Examples include Ultraviolet-visible spectroscopy (UV-VIS), infrared absorption spectroscopy, Raman spectroscopy, nuclear magnetic resonance spectroscopy, and X-ray diffraction spectroscopy [32]. The use of a specific form of spectroscopy or the use of spectral measurement devices (spectrometers or spectrographs) depends on the capability of the sensing system, as well as what medium is being measured or which analyte is attempting to be detected.

#### 2.4.2. Evanescent Field Interaction

The second process is based on the overlap of evanescent waves with a separate medium to measure the refractive index. Figure 4 above shows the propagation of light (indicated by black arrows) as it travels through the core of the fiber and interacts with the cladding and surroundings to form the evanescent field. Most commonly in FOBS, the light propagating through the cladding is insensitive to the surroundings. The evanescent waves decay to approximately zero [9]. A bent (u-shape) fiber optic immunosensor was developed to detect minute levels of human antibody immunoglobulin G (IgG). The small detection limits were achieved through evanescent wave absorbance (EWA) and silver and gold enhancements to the probe. The sharp bend is what mechanically allows for efficient interaction of light and drastic changes in the evanescent field [33].

#### 2.4.3. Photodetectors

Photodetectors sense light and other electromagnetic radiation. Some photodetectors are equipped to convert light photons into a current for signal processing. Two specific types of photodetectors have proven sensitive enough and with an adequate noise level for use in optical fiber sensors. Photomultipliers and semiconductor quantum photodetectors (photodiodes and photoconductors) are commonly used in an application. It has been shown that the use of two photodetectors is more suitable for FOBS because of the need for a reference detector to take in measurements relating to other variables (such as temperature or intensity) [6]. Fluorescent bacteria are being widely researched as sensing agents using optical fibers; however, these applications using bacteria prevent the technologies from being used within the human body or biological fluid due to risk to the patient. A project noted in the journal of coatings technology and research found that coating the optical fiber in glucose oxidase increases the sensing area’s susceptibility to picking up fluorescence signals produced by the organic metal complexes that are quenched by oxygen. This system utilizes the chemical coating as the transducer [27]. In another transducer variation, The Czech Academy of Sciences has developed a FOBS that utilizes magnetic microparticle to determine the presence of biogenic amines. Examples of biogenic amines include histamine and spermine. The magnetic component allows for more efficient immobilization of the proteins [34].

### 2.5. Signal Processors

If the signal is transmitted by a photodetector, it is provided to the signal processor as a voltage or current which is proportional to the intensity of the light signal that was initially measured. Analog circuitry utilizing a current-to-voltage converter or a voltage stage are commonly used options. The signal processor used in FOBS applications is more effective with the incorporation of an amplifier. The signal transmitted through the analog circuitry can then be converted using an analog to digital converter within a computer in order to store and display the data. In some applications, it may be necessary to include electronic background subtraction or noise reduction strategies to improve the output signal [6].

## 3. Discussion

The main components of a FOBS discussed in this paper are optical fiber and transducer, biorecognition element, and photodetectors. Table 1 showcases the main components and the various modifications covered in this paper, as well as, advantages, disadvantages, and applications.

The biosensing elements are what differentiates FOBS from other fiber optic sensors and other biosensors. Enzyme-based FOBS have the best establishment in the market because of their use in glucose monitoring. The ability of enzyme-based FOBS to be miniaturized, and made portable, such as with glucose monitors, gives them the edge over the other sensing elements. Details and more examples were given in Section 2.2.2. Enzymatic sensors also have three distinct mechanisms that can result in signal generation, an advantage that none of the other biosensing elements have. Antibodies as biosensing elements have two options for signal generation. The antibody or antigen can be used to interact with the desired sample to initiate the signal propagation in the fiber and are primarily used with the sensor to develop immunoassays. Both antibody and enzyme-based FOBS are more expensive options than pH or whole-cell based FOBS. pH-based FOBS are the least utilized in the market because of their low variability in applicability. Whole-cell based FOBS are used primarily with fluorescence and bioluminescent cells but are not able to be used in vivo because of the risk of infection or adverse effects.

FOBS can also be differentiated by the sensing strategy utilized. These strategies are all well established in sensing applications and vary based on technology and involved technique. The most commonly used sensing strategies for signal generation are surface plasmon resonance (SPR), fluorescence, interferometers, and refractometers. Each of these strategies takes advantage of a different property of light propagation in the fiber optic cable such as wavelength or angle of refraction to generate the signal. Examples of the various strategies are included throughout the paper but are showcased in Section 2.3 and Section 2.4. Fluorescence is the simplest of the sensing methods to employ, but it does require more equipment and a fluorescent-based signal to produce results [34,35]. The use of refractometers for sensing has the advantage of being versatile and applicable to a range of analytes; however, it does require the design of sensitive layers that change when in contact with the analyte [36]. The use of interferometers has the largest range of characteristics that can be used to analyze the signal including changes in wavelength, phase, intensity, frequency, and bandwidth as sensing indicators. This strategy involves the use of beam splitting and combining components [37]. The signal generated through the strategies mentioned above are then processed through various circuitry.

Although the circuitry itself may vary, the signal is processed and converted through one of two types of signals: voltage or current signals. Both signals can be amplified, converted using analog or digital signal equipment, run through noise reduction and background subtraction apparatuses to enhance the final signal that is output to the interface. Voltage signals are optimized for little noise if the impedance is low and can propagate over larger distances; whereas, current signals are optimized for sending weak signals in a noisy environment.

Fiber optic biosensors are versatile, immune to electromagnetic interference, can be miniaturized, are adaptable to several different industries and are remote operable.

## 4. Future Developments and Industry Trends

Fiber optics for use in biosensors and chemical sensors is well established; however, the versatility of fiber optic-based sensors for in vivo application is becoming more widely successfully researched. In this section, we will present several new developments in this field that most accurately represent the scope of the most recent works. As opposed to developing FOBS for general uses, the trend in the industry seems to be using the already established sensing strategies and a general structure to tailor the sensor to specific applications. Some recent work focuses on applications through integrated systems utilizing FOBS, others are utilizing new modifications to FOBS components, such as the cable itself, or the sensing strategy utilized.

For example, a tapered optical fiber, utilizing a specialized protein (57-mer dopamine-binding aptamer (DBA)) as the biorecognition element has been developed to detect dopamine in vitro. In the presence of dopamine, the protein changes shape and obtains a tertiary structure that changes the refractive index of the fiber surface. This sensor is unique from other dopamine sensors because of its ability to detect dopamine in the presence of other neurotransmitters [39]. In another example, gold nanoparticles were used along with SPR as a sensing strategy and a tilted optical fiber Bragg grating inscribed in standard single-mode fiber to detect thrombin (a protein that aids in the body’s clotting mechanism) up to 1 nM. The fiber optic cable was coating in a 50 nm thick coating of gold. The limit of detection was also enhanced with the addition of 13 nm diameter gold nanoparticles that were bonded to the protein molecules [40]. Proteins are becoming more common industry-wide for use as biorecognition elements because of the wide range of available proteins, the ability of the protein to change the structure in a detectable way when the environment changes, and the biocompatibility of proteins with other biological elements of interest. As chronic diseases become more widely curable and preventable using other medical techniques, sensing systems for rarer, or more specific medical issues, seems to be the most recent trend. This may be due to the high specificity of FOBS and the ability to tailor the sensing system to detect small quantities. Another recent development in thrombin detection has been developed out of the University of Duisburg-Essen in Germany. This system is unique in that it uses a 3-D DNA origami structure as a bioreceptor carrier in combination with SPR with a fiber optic surface. The DNA structure positions thrombin specific aptamers at different distances from the SPR surface. This system boasts better performance in orientation, accessibility, and a wider linear range when compared to other FOBS of a similar type [41]. 

FOBS have several advantages over current conventional analytical sensing systems that make them more appealing to health care and MedTech environments. FOBS do not require electrical connections and, thus, are not affected by electrical interference. This makes them useful in clinical settings because of the plethora of other medical equipment that is present at the time of sampling. This also gives them an advantage over other types of biosensors such as electrochemical biosensors. FOBS have the ability to be somewhat reusable because the biorecognition element does not have to be in direct contact with the optical fibers and can be replaced [42]. Also, as mentioned earlier, FOBS can recognize and have the potential to bind more than one analyte, depending on the application. Although FOBS are more versatile, able to be miniaturized and electrically safer than other biosensors, they do have some disadvantages, such as their sensitivity to ambient light interference and the specificities of the biorecognition elements that require special binding techniques that reduce the effectiveness or speed of the device. 

There is an industry-wide focus on innovation to better predict, treat or cure chronic diseases and ailments. The FOBS research environment is not immune to this trend and several sensing systems have been developed with this aim in mind. For example, Zhou at al also modified a FOBS utilizing an immune molecule as the biosensing agent to detect cardiac biomarkers for the improved diagnosis of myocardial infarction. The sensor was integrated as part of an optical microfiber coupler and detected cardiac troponin I (cTnl). The antibodies were immobilized on the sensing area through a deposition technique. This technology has great potential for clinical use because of its small size and simple detection technique. [43]. The use of FOBS clinically is limited at present; however, the ability of FOBS sensing systems to be specified to areas of general detection and prevention, diagnosing and monitoring gives them value. The continued investment in the development of FOBS will bring them into the clinical atmosphere. It should be noted; however, that FOBS have been in development for decades in the research environment but are limited in their availability to patients. This may be due to the quickened innovation in other areas of medical research that have created a funding funnel that leaves many areas underfunded or perhaps, the operational limitations and variability of FOBS that make them less marketable to large companies and patient groups. FOBS may be more useful in a testing, or lab environment than in a clinical setting with patient interaction.

## Figures and Tables

**Figure 1 diagnostics-09-00023-f001:**
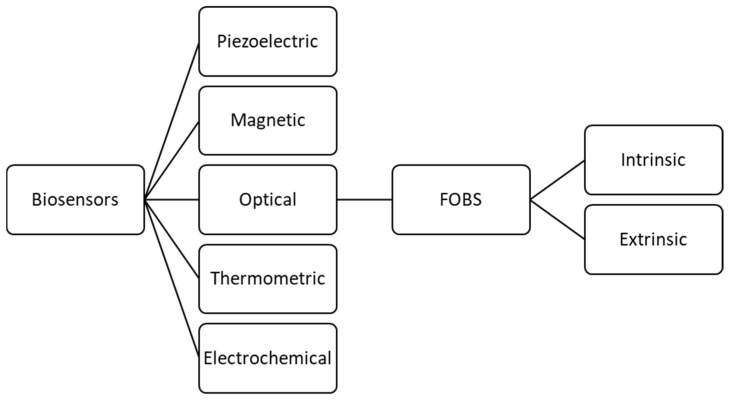
Biosensor breakdown by transducer.

**Figure 2 diagnostics-09-00023-f002:**
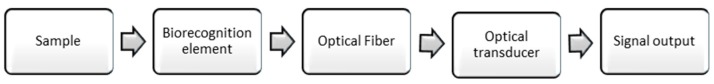
Block diagram of the general flow of the signal of FOBS.

**Figure 3 diagnostics-09-00023-f003:**
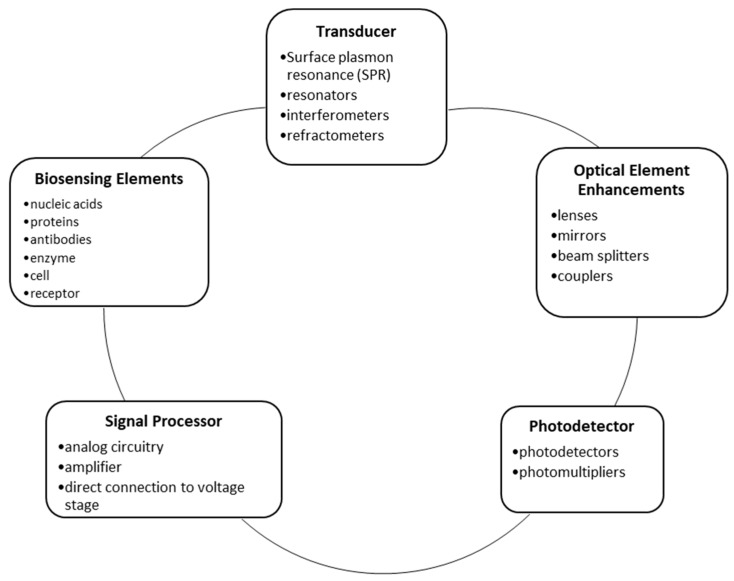
FOBS fundamental architecture.

**Figure 4 diagnostics-09-00023-f004:**
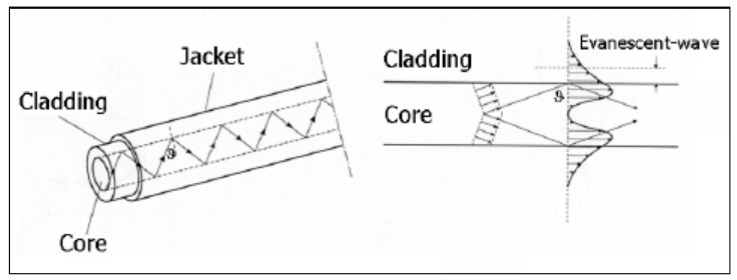
Propagation of light signal in an optical fiber [6].

**Figure 5 diagnostics-09-00023-f005:**
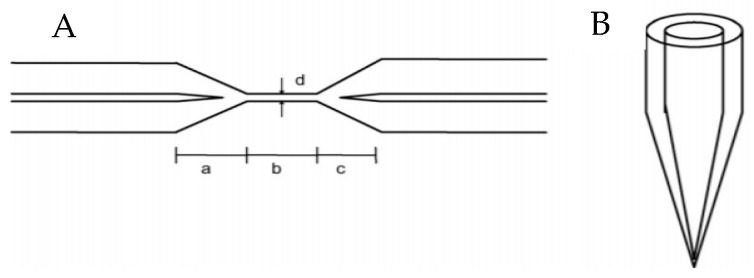
(**A**) Tapered optical fiber (**B**) Tapered tip [11].

**Table 1 diagnostics-09-00023-t001:** Summary of FOBS components and modifications.

Component	Main Modification	Other Options	Application(s)	Advantages	Disadvantages	Reference(s)
**Transducer**	fiber optic cable	material (glass, silica, plastic)	medical, military, telecommunication, industrial, data storage, networking, and broadcast industries	cheaper, more flexible, more durable, easier to manufacture (depending on material type), can be miniaturized	less resistant to damage/unwanted scratching, easier to deform, require replacement faster (depending on material type)	[11,12,13,14]
		core changes (tapering)	sensing	increased evanescent field interaction	---	[11,12,13,14]
		cladding changes (coating, tapering, scraping)	sensing	increased evanescent field interaction, increases sensitivity to specific wavelengths of light	---	[11,12,13,14]
		bending	sensing	increase in the evanescent field	loss of light signal	[11,12,13,14]
		changing the launch angle	sensing	increase evanescent field depth	decreases applicability	[11,12,13,14]
		increasing wavelength	sensing	increase evanescent field depth	---	[11,12,13,14]
**Biosensing element**	extrinsic vs. intrinsic		sensing	extrinsic sensors guide the light to a separate sensing area (ideal for use with biosensors)	intrinsic sensors force the light signal to remain in the core of the fiber to detect changes	[16]
	pH	fluorescence	sensing	gels and other element fixing strategies are replaceable	needs a clear/large enough signal to excite a light generation or fluorescent signal	[9,17]
	enzyme	three options to initiate signal generation	sensing	well established in glucose monitoring, high sensitivity to low oxygen levels in intercellular fluid, is compatible with fluorescent indicators	reduced sensitivity to glucose signal resolution when miniaturized to a microsensor	[10,18,19,20,21,22,23]
	antibody	antigen or antibody use	immunoassay	compatible with fluorescent molecules, wide variety of diagnostic assay capabilities	require an attachment agent in order to be affixed to the cable at the sensing area	[6,24,34]
	whole cell	netting or membranes, fluorescence	bioluminescent cells	use of bioluminescent cells or cells that have measurable metabolic processes, some matrices shown to last up to a month without replacement	not compatible inside the human body because of bacterial infection concerns	[6,26,32]
**Sensing Strategies**	signal generation	surface plasmon resonance (wavelength modulated, flow injection…)	categorized based on detection limit and sample being tested	can detect specific proteins up to small detection limits	---	[29]
		fluorescence	medical applications, chemical sensing and measurements of physical parameters	available in different configurations for specific applications	require more components and a fluorescent based signal in order to produce results	[35]
		interferometers	sensing, telecommunication	measurand can be determined by changes in wavelength, phase, intensity, frequency and bandwidth as sensing indicators	require beam splitting and beam combining components	[36]
		refractometers	sensing, telecommunication	simple, versatile, self-referenced	require design of sensitive layers that change when in contact with the analyte	[35]
**Photodetectors**	spectroscopy	fluorescence and absorption	sensing, telecommunication	well established technology with unlimited variations in technique and applications	have to be sensitive enough and have an adequate noise level to be used with FOBS, require a reference detector	[6,13,27,37]
		evanescent wave interaction	sensing, telecommunication	ability to sense light and other electromagnetic radiation	overlap of evanescent waves with a separate medium to measure the reflective index, requires a reference detector	[5,6,13,27]
**Signal Processor**	voltage signal	amplifier, analog circuitry, analog to digital converter, noise reduction, background subtraction	output to interface, sensors	optimized for little noise if impedance is low, can propagate over larger distances	adds current consumption	[38]
	current signal	amplifier, analog circuitry, analog to digital converter, noise reduction, background subtraction	output to interface, sensors	optimized for sending weak signals in a noisy environment	---	[38]

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
