# Peer review of "A Brief Overview of Medical Fiber Optic Biosensors and Techniques in the Modification for Enhanced Sensing Ability"

_diagnostics, 2019, doi:10.3390/diagnostics9010023_

Round 1
Reviewer 1 Report
This manuscript reviews a selection of research publications concerning the development and application of biosensors that utilise fibre optics for medical applications. The authors begin by providing a lengthy introduction section that define biosensors and how optical fibres have been incorporated into basic biosensor design. They then proceed to review the different modifications made to the fibre optics to impart sensor functionality, summarise key advantages and disadvantages in a summary table with an associated discussion and then look at current industry trends. The authors have significantly improved the structure and flow of the manuscript compared to their earlier submitted version and have a better balance between discussion of fundamental principles of sensor operation and specific examples of different sensor designs. The authors have also added more detail on the industry trends and prognostications on the future direction of the field of research. There are some points for the authors to address and these are detailed below.
1. Page 2 line 77 “. . . for processing [10].” With the erasure of lines 70-73, the reference numbering system will need to be altered to accommodate the removal of the original reference [9] and keep the reference numbering in order throughout the manuscript.
2. Page 5 line 203 Change “. . . has been researched at a low rate.” to “. . . has been little researched.”
3. Page 6 line 232 “. . . acuteness . . .” Do you really mean sensitivity? Clarify your meaning in the manuscript.
4. Page 6 line 238 change “. . . increasing pH intensity, . . .” to “. . . increasing pH, . . .”
5. Page 6 lines 246-247 “There are currently . . . of biological fluids . . .” Can you suggest reasons for this?
6. Page 6 lines 250-255 “For use as . . . the samples binds [10].” Rather than providing one reference to a review article, it would be better to provide specific references for each of the three modes of operation.
7. Page 6 lines 272 “. . . proved too robust . . .” Do you not mean “. . . proved too fragile . . .”?
8. Page 7 lines 281-2 “The system is . . . and phosphate [23].” Xanthine is mentioned twice in this list. Ensure it is only listed once.
9. Page 7 line 318 “. . . more selective in the samples they can bind . . .” Antibodies for instance are typically highly selective binding agents. Do you mean that whole cell organisms have a narrower range of potential analytes that can be detected using them? Clarify this in the manuscript.
10. Section 2.4.2. I think that this section could do with at least one succinct example of an application where evanescent field interaction has been used.
11. Table 1. “. . . shown to last up to a month without” This entry appears to have been prematurely cut-off in the table and so the sentence should be completed.
12. Page 10 lines 425-426 “Antibodies as biosensing . . . for signal generation.” You have not defined clearly what these options are either here or in section 2.2.3. Clarify this in the manuscript.
13. Page 12 lines 512-513 “It should be noted . . . limited in their availability to patients.” Why do you think this is the case? Is this because of some of the operational limitations outlined earlier in the review or are there other reasons for this also? Clarify this in your discussion.
Author Response
On behalf of both authors, we would like to sincerely thank this reviewer for taking the time to read and offer revisions for our paper. The constructive critiques were of great benefit to the growth in skill of the authors and of the quality of the paper. The reviewer’s comments are in italics, followed by the replies.
General Comments:
Thank you for the compliments on the improvements to the paper. The authors are very proud of the way the paper has progressed.
Specific Comments:
1. Reference 9 was deleted from the works cited and the rest of the reference numbers cited in the text were adjusted to accommodate this change.
2. This comment has been implemented.
3. To clarify the characteristic of FOBS being pointed out in this sentence, ‘acuteness’ was replaced with ‘sensitivity’.
4. This change was implemented.
5. Several reasons were proposed by the authors at the conclusion of the sentence.
6. To supplement the review article, an article focused on enzyme biosensors alone was inserted. This article speaks to each of the modes of operation.
7. This comment was addressed in the reviewer’s comments prior to the submission to the diagnostics. Since the comment has reappeared in this reviewer’s comments, the wording has been adjusted.
8. The second listing of xanthine was removed.
9. This comparison was clarified by adjusting the wording of this section and adding more detail.
10. An example was inserted to keep with the format of the paper and to illustrate the concept.
11. This revision has been made.
12. This point was clarified and enhanced with additional detail in section 2.2.3.
13. This was clarified.
I can recomend this paper to be pubish in the revised vesion
Author Response
On behalf of both authors, we would like to sincerely thank this reviewer for taking the time to read and offer revisions for our paper. The constructive critiques were of great benefit to the growth in skill of the authors and of the quality of the paper. The reviewer’s comments are in italics, followed by the replies.
General Comments:
Thank you for the time you spent reading the revised paper and for your recommendation for publication. No revisions were made as a result of reviewer #2.Reviewer 3 Report
The manuscript has been improved, but the term "systematic" in the title and in the abstract is misleading and makes me expect a deeper insight into the different topics and at least three times the references reported for such a broad topic. I think "brief overview" could be more appropriate than "systematic review".
Author Response
On behalf of both authors, we would like to sincerely thank this reviewer for taking the time to read and offer revisions for our paper. The constructive critiques were of great benefit to the growth in skill of the authors and of the quality of the paper. The reviewer’s comments are in italics, followed by the replies.
General Comments:
At the suggestion of the reviewer, the word “systematic” was replaced by “brief overview” in both the paper title and abstract.This manuscript is a resubmission of an earlier submission. The following is a list of the peer review reports and author responses from that submission.
Round 1
Reviewer 1 Report
This manuscript sets out to review the combination of fibre optic technology with biosensor transduction as used in medical applications. Whilst a review of the most recent advances in fibre optic biosensing in medicine is of clear interest to medical technologists and analytical biochemists, I feel that the current manuscript has a number of deficiencies, most notably:
A. There is too much focus on extremely fundamental aspects of the technology (e.g. defining a biosensor and the steps in biosensor functioning) and not enough focus on the actual medical applications and the performance of the devices in a clinical setting.
B. The review lacks consistent structure with fundamental theoretical principles repeated across different sections, a separate discussion section that provides little or no additional insight on and above section 2, and a conclusions section that lacks any prognostications on the future trajectory of this research.
C. The review contains a lot of vague and generalised statements that don’t add much value to the discussion.
D. The standard of spelling, grammar and syntax is poor. The manuscript needs to be completely re-edited.
Additional more specific points for the authors are given below.
1. Page 1 lines 21-23 “The need for . . . less invasive.” This is a very generalised statement. There will always be a demand for effective, accurate, mobile and non-invasive sensing irrespective of demographic trends. You are not clearly showing here how much fibre-optic based sensors can contribute to these desirable sensor properties.
2. Introduction. A great deal of space is devoted to very fundamental and simplistic definition of biosensors and fibre optic biosensors as a subset of this. The authors could reasonably assume much of the material in the introduction is already understood by the reader. Focussing this section on the key underpinning concepts in fibre optic design would enable more expansive description of the latest developments in the field.
3. Page 4 lines 168-170 “To test this . . . of cholesterol [22].” It would be useful to include succinct details on how the sensor overcame interferences from the blood serum, i.e. by what mechanism it is a specific sensor.
4. Page 5 lines 190-191 “The first component . . . the biosensing element . . .” This is a poorly worded statement. The biosensing element is the source of the fundamental signal or change in signal, the detection of which is achieved by the biosensor as a whole.
5. Page 5 lines 207-208 “The results showed . . . a linear fashion.” This is better worded as a pH increase resulting in a fluorescence increase (i.e. fluorescence is the dependent variable).
6. Page 5 line 218 “. . . or individual.” It is not clear what you mean here. What is an “individual” in this context? Clarify this in the manuscript.
7. Page 5 line 236 “. . . University of Maryland School of Maryland . . .” should read “. . . University of Maryland School of Medicine . . .”
8. Page 6 lines 238-239 “This technique proved . . . biological environment [11].” Do you not mean “too fragile” rather than “too robust”? Clarify your meaning in the manuscript.
9. Page 6 lines 253-255 “Similar to . . . carboxyl groups [6].” Whilst antibodies are typically immobilised in this way, they can also be immobilised without these moieties. However, the key issue is orientation of the antibody relative to the fluid phase containing the antigen.
10. Page 6 line 260 “ . . . with sensitivity up to 10-11M.” What is the target compound for this assay? Include details in the manuscript. Also, the detection limits should not be 10-11 M which is a very high concentration – revise the units used.
11. Page 6 line 265 “. . . levels of PC . . .” All abbreviations should be defined at first use in the manuscript.
12. Page 6 line 275 “. . . broader degree of selectivity . . .” It is not clear what you mean here. Do you mean they have lower selectivity? Clarify this in the manuscript.
13. Page 6 line 278 “. . . androgenic compounds, androgens . . .” you should only list “. . . androgens .. .” here.
14. Page 7 lines 286-296 “FOBS utilizing . . . interferometer sensor [27].” This section is poorly structured. You offer a selection of generalised comments on SPR with no examples and then in the same paragraph suddenly jump to talking about interferometers. Expand on each transduction method more thoroughly, with examples, in separate paragraphs.
15. Page 7 lines 290-296 – Use of reference [27]. This reference is a review article. Rather than reviewing a review article, it is better to use a primary research paper to support these statements.
16. Page 7 lines 303-304 “The evanescent . . . electromagnetic radiation.” Once again, you are jumping suddenly from one concept (evanescent waves) to another (photodetectors) without making an obvious connection between the two. This section should be restructured to separately describe each concept in turn.
17. Page 7 line 333 - page 8 line 339 “Sensing strategies . . . cable itself.” This text is merely going over the same ground you covered earlier in the review and so is needless repetition.
18. Table 1. “. . . is less ideal if the signal is slow . . .” (under voltage signal disadvantage) – why is it less ideal under these conditions?
19. Page 9 lines 357-367 “As mentioned in . . . be more expensive.” This is more repetition of earlier content and is not really needed.
20. Page 9 lines 370-371 “Miniaturization and portability . . . other sensing elements.” You haven’t really covered miniaturization and portability in any detail and not in comparison to the options available with other biosensor technologies. Add more detail in the manuscript on how fibre optic biosensors allow portability and miniaturization over and above other options.
21. Page 10 lines 412-415 “For example, . . . decibels [36].” You need to provide details in the manuscript of how this technology is an example of a biosensor.
22. Page 10 lines 412-430 “For example, . . . detection technique [38].” These examples should be in the earlier discussion sections of the review not in the conclusions section. Instead the space should be used to draw useful conclusions on the state-of-the-art of medical fibre optic biosensors and their future prospects.
Reviewer 2 Report
The manuscript “A Systematic Review of Medical Fiber Optic Biosensors and Techniques in Modification for Enhanced Sensing Ability”, by S. Mowbray & A. Amir presents several examples of fiber optic biosensors.
The authors list biorecognition elements in order to explain main difference between fibre optic sensor and fiber optic biosensors.
I may recommend this review to be published in the Health journal after some modification.
1) As Title of the manuscript introduces "Medical sensors", so examples of medical applications should be given. Examples will enable to illustrate how FOBS responds to medical need.
2) All biorecognition elements (antibody, whole cell, enzymes…) can be used in other types of biosensors (for instance electrochemical). What are advantages when FO are used? How fiber optic improves in sensing?
Reviewer 3 Report
The manuscript by S. Mowbray and A. M. Amiri reviews fiber optic technologies for biomedical applications. The manuscript is poorly organized, and the authors jump from one concept to the other without explaining properly any of them. The manuscript is then hardly readable, and I do not think it can provide a relevant contribution to the field. Therefore, I do not recommend it for publication in Healthcare.
Beside my general comment, the manuscript presents the following issues (among many others):
Figures are not properly commented in the text
References are not in numerical order in the text, and the number of references is not appropriate for a review paper
Table 1 is completely unreadable